# *UMOD* Mutations in Chronic Kidney Disease in Taiwan

**DOI:** 10.3390/biomedicines10092265

**Published:** 2022-09-13

**Authors:** Huan-Da Chen, Chih-Chuan Yu, I-Hsiao Yang, Chi-Chih Hung, Mei-Chuan Kuo, Der-Cherng Tarng, Jer-Ming Chang, Daw-Yang Hwang

**Affiliations:** 1Department of Pathology, Kaohsiung Medical University Hospital, Kaohsiung Medical University, Kaohsiung 807377, Taiwan; 2National Institute of Cancer Research, National Health Research Institutes, Tainan 70456, Taiwan; 3Department of Laboratory Medicine, Kaohsiung Medical University Hospital, Kaohsiung Medical University, Kaohsiung 807377, Taiwan; 4Department of Medical Imaging, Kaohsiung Medical University Hospital, Kaohsiung Medical University, Kaohsiung 807377, Taiwan; 5Division of Nephrology, Department of Internal Medicine, Kaohsiung Medical University Hospital, Kaohsiung Medical University, Kaohsiung 807377, Taiwan; 6Institutes of Physiology and Clinical Medicine, Division of Nephrology, Department of Medicine, Taipei Veterans General Hospital, National Yang-Ming Chiao-Tung University, Taipei 112201, Taiwan; 7Center for Biomarkers and Biotech Drugs, Department of Biomedical Science and Environmental Biology, Kaohsiung Medical University, Kaohsiung 807377, Taiwan

**Keywords:** uromodulin, chronic kidney disease (CKD), autosomal dominant tubulointerstitial kidney disease (ADTKD), kidney failure (KF)

## Abstract

*UMOD* is the first identified and the most commonly mutated gene that causes autosomal dominant tubulointerstitial kidney disease (ADTKD). Recent studies have shown that ADTKD-*UMOD* is a relatively common cause of chronic kidney disease (CKD). However, the status of ADTKD-*UMOD* in Taiwan remains unknown. In this study, we identified three heterozygous *UMOD* missense variants, c.121T > C (p.Cys41Arg), c.179G > A (p.Gly60Asp), and c.817G > T (p.Val273Phe), in a total of 221 selected CKD families (1.36%). Two of these missense variants, p.Cys41Arg and p.Gly60Asp, have not been reported previously. In vitro studies showed that both uromodulin variants have defects in cell membrane trafficking and excretion to the culture medium. The structure model predicted altered disulfide bond formation in both variants, but only p.Gly60Asp was predicted to cause protein destabilization. Our findings extend the mutation spectrum and indicate that the ADTKD-*UMOD* contributed to a small but significant cause of CKD in the Taiwanese population.

## 1. Introduction

ADTKD is a rare dominant inheritance disease caused by heterozygous mutations in *UMOD* [1], *MUC1* [2], *HNF1B* [3], *REN* [4], and *SEC61A1* [5]. *UMOD* is the first identified and the most common cause of ADTKD, and it has been subject to extensive investigation [1,6]. Although rare, ADTKD is the third most common monogenic kidney disease following ADPKD and collagen type IV nephropathy, but the prevalence of ADTKD-*UMOD* remains unclear, and recent studies show that it represents up to 2% of patients with KF or 0.3% of all individuals with CKD [7,8].

*UMOD* encodes uromodulin and is the most abundant protein in human urine [9]. Uromodulin is a kidney-specific protein produced by the epithelial cells of the thick ascending limb of the loop of Henle and the early part of the distal convoluted tubule [10], and it is formed as a high-molecular-weight polymer via its zona pellucida domain [11]. Moreover, uromodulin is a highly glycosylated protein, with seven *N*-glycosylation sites and cleavage at the end of the C-terminal by serine protease hepsin, before being excreted into the tubular lumen [9,12,13]. The function of uromodulin includes protection against calcium oxalate crystallization [14], defense against infections in the urinary tract from uropathogenic bacteria [15,16,17,18], and regulating the Ca^2+^-K^+^ ion channel [19]. The effect of *UMOD* mutations is delayed maturation, retention in the endoplasmic reticulum (ER), reduced expression at the plasma membrane, and decreased urinary excretion, which lead to ER stress and the unfolded protein response [20,21,22].

Clinical phenotypes of ADTKD-*UMOD* are nonspecific and include CKD, gout, hyperuricemia, normal to mild proteinuria, presence or lack of a family history, and most affected individuals entering KF between the ages of 30 years and 50 years [23]. Renal histology shows tubular dilation or atrophy, tubular basal membrane lamellation or thickening, interstitial fibrosis, and accumulation of mutated uromodulin as polymorphic unstructured materials revealed by PAS staining [24,25].

Taiwan has the highest incidence and prevalence of CKD and KF based on international comparisons of the USRDS report [26]. However, no study has focused on the status of ADTKD-*UMOD* in the CKD population in Taiwan. In this study, we examined the prevalence of ADTKD-*UMOD* mutations in a selected CKD cohort in a single tertiary center in Taiwan.

## 2. Materials and Methods

### 2.1. CKD Individuals and DNA Preparation

This study recruited 221 CKD families (228 individuals) from the CKD Care Program of Kaohsiung Medical University Hospital. The inclusion criteria included gout episodes or hyperuricemia with eGFR less than 90 mL/min/1.73 m^2^ before the age of 40. Blood samples, pedigree information, and access to results of laboratory work were obtained from individuals after informed consent was given. The study was conducted following the Declaration of Helsinki and approved by the Institutional Review Board of KMUH (KMUHIRB-G(II)-20160024). DNA from peripheral blood samples was extracted according to the standard method.

### 2.2. Exome Sequencing and Bioinformatics Analysis

The Nextera Flex for Enrichment and Exome panel (CEX V2, Illumina, San Diego, CA, USA) was used for exome library creation. Size selection, sequence capture, enrichment, and elution were performed according to the manufacturer’s instructions. The size distribution of DNA libraries was then measured using TapeStation 4150 (Agilent, Santa Clara, CA, USA), followed by sequencing on an Illumina NovaSeq 6000 with 150 bp paired-end reads. The resulting fastq data were aligned to the reference human genome sequence (GRCh37/hg19) and we performed nucleotide variant calling using the DRAGEN Bio-IT platform (Illumina). The VCF file was analyzed in CLC Genomics Workbench (Qiagen, Germantown, MD, USA). Identified variants were labeled as pathogenic/likely pathogenic, variant of uncertain significance (VUS), or benign, according to the American College of Medical Genetics and Genomics (ACMG) classification, using the VarSome software, and we compared variants that existed in the ClinVar, HGMD, and dbSNP databases. Detected pathogenic, likely pathogenic, and VUS variants were confirmed by Sanger sequencing. For Sanger sequencing, the PCR products were purified by shrimp alkaline phosphatase/exonuclease I (78390, USB Products, Affymetrix, Santa Clara, CA, USA) treatment and then subjected to sequencing from both ends by thermocycler sequencing using the BigDye^®^ terminator 3.1 sequencing kit (4337458, Applied Biosystems, Thermo Fisher Scientific, Waltham, MA, USA) following analysis on an ABI 3730XL DNA Analyzer (Applied Biosystems, Thermo Fisher Scientific). Segregation analysis was performed if DNA was available from family members. Exons 2, 3, 4, and 5 of *UMOD* were screened by Sanger sequencing in the proband of 221 CKD families except the DY5 family, which received exome sequencing. The NCBI Ref sequence NM_003361.4 is used for *UMOD*. The standard nomenclature recommended by the Human Genome Variation Society (http://www.hgvs.org/; accessed on 18 July 2022.) was used to number nucleotides and name mutations or variants.

### 2.3. Cell Culture and Transient Transfection

HEK293 cells (CRL-1573, ATCC, Manassas, VA, USA) were cultured in Kaighn’s modification of Ham’s F-12 medium (F-12K) (N3520, Merck/Millipore Sigma, Burlington, MA, USA) supplemented with 10% heat-inactivated fetal bovine serum (10437028, Gibco, Thermo Fisher Scientific, Waltham, MA, USA), 100 units/mL penicillin, 100 μg/mL streptomycin (10378016, Gibco, Thermo Fisher Scientific), and 10 units/mL γ-interferon (IF002, Merck/Millipore Sigma). The cells were maintained at 37 °C in a humidified atmosphere of 5% CO_2_.

Transient transfection of HEK293 cells was performed using Lipofectamine 2000 (11668027, Invitrogen, Thermo Fisher Scientific, Waltham, MA, USA). Briefly, 2.5 μL of Lipofectamine 2000 was added directly into 100 μL Optimum medium (31985062, Gibco, Thermo Fisher Scientific) and incubated for 5 min at room temperature. A total of 1 μg of DNA was added to the Lipofectamine 2000–Optimum mixture and incubated for another 20 min. The DNA mixture was added to cell culture plates with a serum-free medium dropwise at the end of incubation. For the health status of the cells after transient transfection, we used the LUNA-II™ Automated Cell Counter (Aligned Genetics, Anyang-si, Gyeonggi-do, Korea) to detect trypan blue staining cell viability at a different time point, and most transfected cells remained alive upon different uromodulin over-expression (data not shown).

### 2.4. Site-Directed Mutagenesis

The HA-tagged wildtype uromodulin vector was obtained from Dr. Rampoldi [27]. The mutation constructs of p.Cys41Arg and p.Gly60Asp were created using the QuikChange II Site-Directed Mutagenesis Kit (200523, Agilent) according to the manufacturer’s instructions. The mutations were confirmed by Sanger sequencing.

### 2.5. ELISA and Fractionation

Empty vector, wildtype uromodulin, p.Cys41Arg uromodulin, and p.Gly60Asp uromodulin expression constructs were transiently transfected to HEK293 cells. After transient transfection, the culture medium was collected at 8, 24, and 32 h, then stored at −80 °C before measurement. For immunoblotting, the culture medium was condensed by Centricon (YM3, amicon, Merck/Millipore Sigma) before SDS-PAGE electrophoresis. ELISA was used to carry out measurements of the uromodulin in the culture medium according to the manufacturer’s instructions (EHUMOD, Invitrogen, Thermo Fisher Scientific). Briefly, the culture medium and standards were diluted in an assay diluent supplied in the kit and added to wells in duplicate. After 2.5 h of incubation at room temperature, the wells were washed with PBS, and the biotinylated uromodulin antibody was added. After 1 h of incubation at room temperature, the wells were rewashed, and a streptavidin-horseradish peroxidase secondary antibody was added. After 45 min of incubation at room temperature, the wells were rewashed, and the color was developed by adding TMB substrate solution and incubation in the dark at room temperature for 30 min. The reaction was stopped by adding the stop solution supplied in the kit, followed by reading immediately at OD450 and OD550. The culture medium’s uromodulin concentration was determined by interpolation on the standard curve.

After the culture medium was collected, the cells attached to the culture dish proceeded to fractionation using a subcellular protein fractionation kit (78840, Thermo Fisher Scientific) according to the manufacturer’s instructions. In brief, we collected equal amounts of three different types of uromodulin cell lysate and removed all the supernatant until as dry as possible. We added ice-cold cytoplasmic extraction buffer containing protease inhibitors to the cell pellet, incubated at 4 °C for 10 min with gentle mixing, then centrifuged at 500× *g* for 5 min. We immediately removed the supernatant (cytoplasmic extract) to a clean pre-chilled tube on ice and added an ice-cold membrane extraction buffer containing protease inhibitors to the pellet. The membrane portion was vortexed for 5 s on the highest setting and incubated at 4 °C for 10 min with gentle mixing. We transferred the supernatant (membrane extract) to a clean pre-chilled tube on ice after centrifuging at 3000× *g* for 5 min. The subcellular membrane and cytoplasmic protein were analyzed by western blotting.

### 2.6. Immunoblotting Analysis

The protein samples were gathered from the cells with an RIPA buffer (R0278, Merck/Millipore Sigma) or the fractionation step. Subsequently, the protein concentration was determined using a protein assay dye (5000006, BIO-RAD, Hercules, CA, USA) and access to an equal amount of protein, then separated by electrophoresis on SDS-PAGE. After the proteins were transferred on polyvinylidene difluoride (PVDF) membranes (1620177, BIO-RAD), the PVDF membrane was immersed in 5% skim milk at 1X TBST, blocking for 1 h at room temperature, and washed three times with 1X TBST for 5 min. Next, we incubated the PVDF membrane with the primary antibody at 4 °C overnight, washed it three times with 1X TBST for 5 min, and incubated it with the secondary antibody for 1 h at room temperature. It was washed three times with 1X TBST for 5 min after secondary antibody incubation, then we added the enhanced chemiluminescence (ECL) substrate kit (GERPN2134, Merck/Millipore Sigma) and detected the protein signal using an X-ray processor (MXP-101, KODAK, Rochester, NY, USA) on X-ray film (Super RX, FUJIFILM, Minato-ku, TKY, JPN). Primary and secondary antibodies were diluted with 5% skim milk in 1X TBST, and the detailed information isas follows: 1:2000 for uromodulin (sc-20631, Santa Cruz, Dallas, TX, USA) and 1:20,000 for rabbit HRP 2nd antibody (GTX213110-01, GeneTex, Irvine, CA, USA).

### 2.7. Protein Structure Prediction

Dynamut was applied for the 3D structure prediction of the wildtype and mutant uromodulins (http://biosig.unimelb.edu.au/dynamut/; accessed on 15 June 2022.) using the full-length structure of the native human uromodulin (Protein Data Bank: 7PFP) [28]. DiANNA was used for the disulfide bond prediction (http://clavius.bc.edu/~clotelab/DiANNA/; accessed on 16 June 2022.) of the wildtype and mutant uromodulins [29,30,31].

### 2.8. Statistic Analysis

ImageJ analysis software analyzed and quantified the western blot bands. The statistical analyses and graphs were carried out and prepared using GraphPad Prism8 software (GraphPad Software Incorporation, San Diego, CA, USA), and the mean ± SEM was presented. An unpaired t-test was applied for a two-group comparison. The statistical results were marked as follows: * *p* < 0.05; ** *p* < 0.01; *** *p* < 0.001. Each experiment was repeated at least three times.

## 3. Results

The study comprised 221 families (228 individuals) where the index patients met the inclusion criteria of gout, hyperuricemia, and an eGFR of less than 90 mL/min/1.73 m^2^ before the age of 40 years old from a single medical center in southern Taiwan. Exon screening of the *UMOD* gene in a total of 220 families and exome sequencing in a single DY5 family (three affected members) were performed. Identified variants went through ACMG classification with 28 criteria to evaluate their pathogenicity, followed by family segregation analysis along with population database (gnomAD and TOPMed) and clinical database (ClinVar) comparisons. Finally, a total of three heterozygous *UMOD* missense variants, c.121T > C (p.Cys41Arg), c.179G > A (p.Gly60Asp), and c.817G > T (p.Val273Phe), were identified, which accounted for 1.36% of this cohort (Table 1). The p.Val273Phe variant was reported twice in the ClinVar database and classified as likely pathogenic; thus, no further analysis was performed. Pedigree, clinical presentation and imaging, sequencing analysis, amino acid conservation, in vitro functional studies, and mutant protein and disulfide bond prediction were performed to evaluate the pathogenicity of the p.Cys41Arg and p.Gly60Asp variants.

Family DY5: Exome sequencing was performed in DY5 III-1, III-2, and III-4. A total of 28 variants, including *UMOD* c.179G > A (p.Gly60Asp), were shared among these three affected individuals after exome sequencing analysis (Appendix A). Bioinformatic analysis showed that only c.179G > A (p.Gly60Asp) was classified as VUS by the ACMG classification and as likely pathogenic by the Varsome software. This variant does not exist in the 1000 Genomes, gnomAD, TOPMed, and ClinVar databases. Furthermore, this variant fits the family’s dominant inheritance pattern and is segregated from the phenotype of all the family members, indicating that this variant may be the disease-causing mutation of the DY5 family (Table 1 and Figure 1a). The affected members had hyperuricemia but no gout episode and entered KF between the ages of 26 and 41 (II-1 34 y/o, III-1 26 y/o, III-2 30 y/o, III-3, III-4 41 y/o). In DY5 III-2, his kidney function deteriorated rapidly, with a declining annual rate of more than 9 mL/min/1.73 m^2^ between the age of 22 and 30 (Figure 2a). Magnetic resonance imaging at the age of 29 showed an irregular kidney contour with tiny bilateral cysts (Figure 2b).

Family CKD401 and WG: Two *UMOD* variants were identified in index individuals with gout or hyperuricemia before the age of 40. The *UMOD* variant c.817G > T (p.Val273Phe) was identified in individual CKD401; he presented with hyperuricemia and gout episodes and received renal replacement therapy in his late fifth decade and had an unknown family history (Table 1). In the WG family, the *UMOD* variant c.121T > C (p.Cys41Arg) was segregated in the affected members and was likely pathogenic according to the ACMG classification. Members II-1 and II-2 of the WG family experienced hyperuricemia and acute gout episodes from adolescence, with CKD stage 3 in their early third decade. The I-1 mother received renal replacement therapy in her early fifth decade. (Table 1 and Figure 1b). Both *UMOD* p.Gly60Asp and p.Cys41Arg do not exist in the known databases and have not been reported before, but they appear to be highly conservative across different species, except chicken and zebrafish, in the UCSC Genome Browser (Figure 1c), and their functional effects have been tested in vitro.

To determine the effect of uromodulin c.179G > A (p.Gly60Asp) and c.121T > C (p.Cys41Arg), two mutant constructs were made by using site-directed mutagenesis. After 24 hr of transient transfection with an equal number of HEK293 cells, a similar amount of uromodulin protein can be identified in the whole-cell lysates of WT and mutant-transfected cells (Figure 3a). A faint band can be seen in the cell transfected with an empty vector, indicating the existence of endogenous uromodulin in the HEK293 cells. Uromodulin excretion from cells was tested by collecting the culture medium after 8, 24, 32, and 48 h of transient transfection, followed by ELISA measurement. Wildtype transfected cells had a higher uromodulin content in the culture medium (6 ng/mL at 48 h) compared with the two mutants (3 ng/mL at 48 h) (Figure 3b), and we also observed the same improper uromodulin excretion by western blotting (Figure 3a), indicating that the p.Gly60Asp and p.Cys41Arg variants affect uromodulin excretion.

Membrane portions of WT, p.Gly60Asp, and p.Cys41Arg transfected cells were collected 24 h after transfection. The western blot showed that only WT uromodulin had influential bands around 100kDa, and only faint signals were detected in the p.Gly60Asp and p.Cys41Arg uromodulin membrane portions (Figure 3c). In contrast, the cytosol portion showed a similar uromodulin expression pattern among all three groups (Figure 3d). These results indicate that most mutant uromodulin has a defect in trafficking to the cell membrane and is most likely retained in the cytosol.

The analysis of uromodulin protein structure by DynaMut showed that the p.Cys41Arg variant changes the Gibbs free energy (DDG) to 0.018 kcal/mol with the same number of hydrogen bonds, which may not significantly change its stability (Figure 4a). However, p.Gly60Asp was predicted to have a DDG of −0.423 kcal/mol and hydrogen bonds different to those of the WT, which may cause protein destabilization (Figure 4b). Predictions of the disulfide bonds of WT, p.Cys41Arg, and p.Gly60Asp were calculated using DiANNA. A total of 24 disulfide bonds were formed by 48 conserved cysteine residues in the wildtype uromodulin. The prediction model showed that multiple disulfide bonds might be affected by p.Cys41Arg and p.Gly60Asp variants (Figure 4c), although p.Gly60Asp does not directly affect the disulfide bridges.

In summary, our functional experimental results support that both the p.Cys41Arg and p.Gly60Asp variants are disease-causing *UMOD* mutations exhibiting delayed protein trafficking to the cell membrane, increased cytosol accumulation, and decreased extracellular excretion, and they affect the protein disulfide bond formations.

## 4. Discussion

Our result shows that ADTKD-*UMOD* represented a small but significant 1.36% (3/221) of the young CKD population with hyperuricemia or gout in Taiwan. This result is compatible with that of previous studies where ADTKD-*UMOD* constituted around 0.3% of all CKD individuals [7,8,32]. No previous publications have a molecular diagnosis of ADTKD-*UMOD*, which indicates that this is an under-diagnosed disease entity because genetic diagnosis is not widely available and not covered by health insurance in Taiwan. Clinical genetics are essential in proteomics, clinical phenotype, digital pathology, and clinical outcomes in precision medicine [33].

Several large-scale ADTKD-*UMOD* genetic analyses have been performed in recent years [6,34,35], and most mutations are located in exons 3 to 5 [36,37,38,39]. All three variants in our cohort are also located in exon 3, but in different domains. Both p.Gly60Asp and p.Cys41Arg were in the EGFR-like domain, and p.Val273Phe was in the D8C domain and has been previously reported [40]. Two other variants had not been previously reported or did not exist in any databases, but a similar p.Cys41Tyr has been reported previously [6]. These two novel variants are segregated in their families, do not exist in any available public databases (gnomAD and TOPMed), clinically fit the known ADTKD-*UMOD* phenotype, and are classified as likely pathogenic (p.Cys41Arg) and VUS (p.Gly60Asp) according to the ACMG criteria. Functional in vitro studies showed delayed uromodulin maturation to the cell membrane and decreased uromodulin secretion of these two variants, further supporting their pathogenicity as the disease-causing mutation in these two families.

The signal peptide of uromodulin is first cleaved in the endoplasmic reticulum with *N*-glycosylation, disulfide bridge formation, and glypiation of the C-terminus, followed by further modification in the Golgi apparatus and glycosylphosphatidylinositol-anchored to the apical membrane [41]. The membrane uromodulin is actively released in the urine by proteolytic cleavage at residue Phe587 [13]. Normal uromodulin in the urine provides several functions, including protection against bacterial infection, maintaining innate immunity, protection against kidney stone formation, and even blood pressure control [14,15,16,17,18,42]. Our in vitro studies showed that these mutants had a decreased uromodulin content in the culture medium and a decreased uromodulin content in the cell membrane portion, indicating that these mutants’ uromodulin was most likely retained in the cytosol. Our results are compatible with those of previous functional studies where retained mutant uromodulin was found to lead to maturation defects in Golgi apparatus, trafficking to the cell membrane followed by ER stress, tubular cell damage, and clinical CKD or KF [20,21,40,43,44].

Amino acid mutations might cause protein structure and residue interaction changes. Regarding uromodulin’s protein structure, the cysteine at position 41 (Cys41) is one of the 48 conserved cysteine residues involved in the formation of 24 disulfide bridges [45]. Cys41 formed a disulfide bridge with Cys126, and mutation at Cys126 has been reported [46]. The p.Cys41Arg variant can change the disulfide bond formation and lead to improper protein folding and the export of the precursor protein from the ER [47]; even p.Gly60Asp may indirectly affect the disulfide bond formation according to the DiAANA prediction. Male gender and high in vitro mutation scores according to the measurement of the severity of the trafficking defect of the uromodulin mutants were significant predictors of worse kidney outcomes in ADTKD-*UMOD* [34]. We classified the p.Cys41Arg and p.Gly60Asp mutants as having a high mutation score of 4 based on the mature uromodulin expression being barely detected by the Western blot. These results are compatible with our clinical observation that all family members of p.Cys41Arg and p.Gly60Asp entered KF at a relatively young age. However, we did not find that the incidence of gout at a younger age was associated with a younger age of KF, since all the affected DY5 individuals never had gout.

Several limitations existed in our study. First, not all *UMOD* exons were included in the mutation screening, and mutations in the exons not screened may have been missed. Second, our method could not detect large indel mutations or complex gene rearrangement. Furthermore, other ADTKD-related genes, including *MUC1*, *REN*, *HNF1B*, and *SEC61A1*, were not included in this study.

## 5. Conlusions

In conclusion, we identified that *UMOD* mutations were present in a small but significant number of young individuals with CKD in Taiwan. Furthermore, we provided evidence that these two novel *UMOD* variants are disease-causing mutations in these two families through functional studies and a protein structure analysis. Multi-center screening should provide a better mutation landscape in Taiwan and contribute to the ADTKD research field.

## Figures and Tables

**Figure 1 biomedicines-10-02265-f001:**
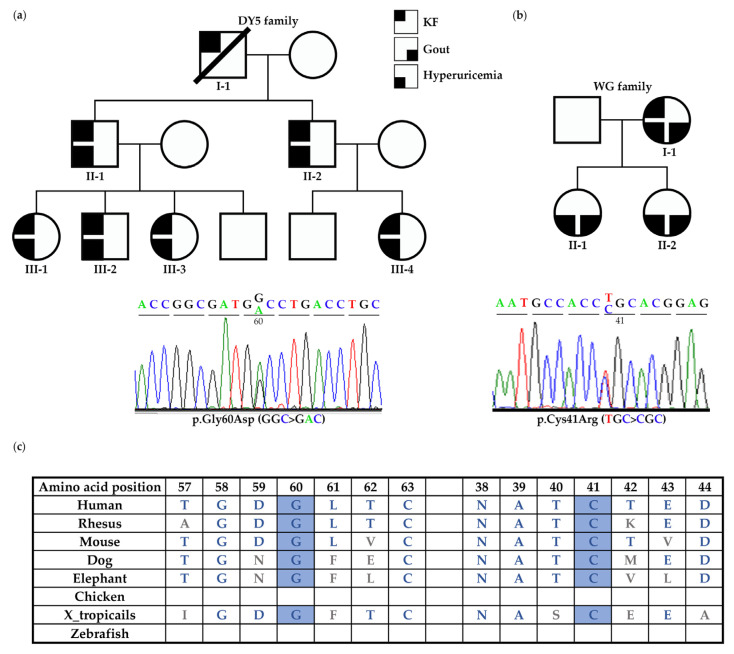
Pedigree and Sanger sequencing electropherogram showing *UMOD* variants in DY5 and WG families. (**a**) In the DY5 family, the *UMOD* p.Gly60Asp variant was shared among all the affected individuals. All the affected members had hyperuricemia but no gout episode and entered KF between the ages of 26 and 41. (**b**) The *UMOD* p.Cys41Arg variant was identified in the WG family. The affected members had gout, hyperuricemia, and CKD or KF. (**c**) The amino acids of Cys41 and Gly60 are well-conserved across different species.

**Figure 2 biomedicines-10-02265-f002:**
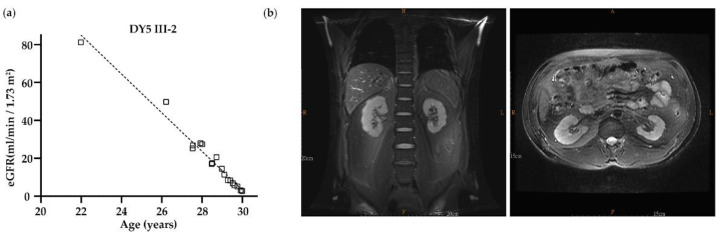
The clinical data from DY5 III-2. (**a**) Age versus estimated glomerular filtration rate (eGFR); the correlation between age and eGFR was analyzed by linear regression. The regression line was included, and eGFR was significantly decreased while aging. (**b**) MRI at the age of 29 showed that the contour of the bilateral kidneys was lobulated with some small cysts in the medullary pyramid, while the left kidney was smaller in size and thinner in the cortex compared to the right.

**Figure 3 biomedicines-10-02265-f003:**
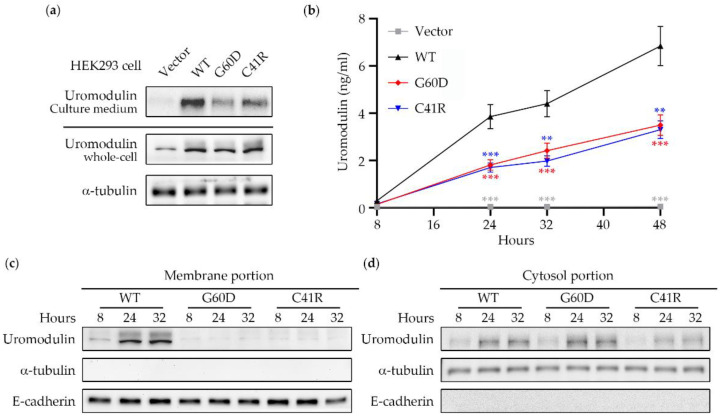
Uromodulin expression profile in transient-transfected HEK293 cells. (**a**) After 24 h of transient transfection with an equal number of HEK293 cells, a similar amount of uromodulin protein could be identified in the whole-cell lysates of WT and mutant-transfected cells. The uromodulin portion condensed from the culture medium was higher in WT compared to the other two groups. (**b**) Wildtype transfected cells had a higher uromodulin content in the culture medium (6 ng/mL at 48 h) compared with the two mutants (3 ng/mL at 48 h). The statistic results were presented as ** (*p* < 0.01) and *** (*p* < 0.001). (**c**) Membrane portions and (**d**) cytosol portions of WT, G60D, and C41R transfected cells were collected after 24hr after transient transfection. The western blot showed that WT uromodulin had influential bands around 100kDa, and only faint signals were detected in the G60D and C41R uromodulin membrane portions. In contrast, uromodulin gradually increased over time and showed a similar expression pattern among the three groups. E-cadherin is represented as a membrane marker, while a-tubulin is a cytosol marker. WT: wildtype; G60D: p.Gly60Asp; C41R: p.Cys41Arg.

**Figure 4 biomedicines-10-02265-f004:**
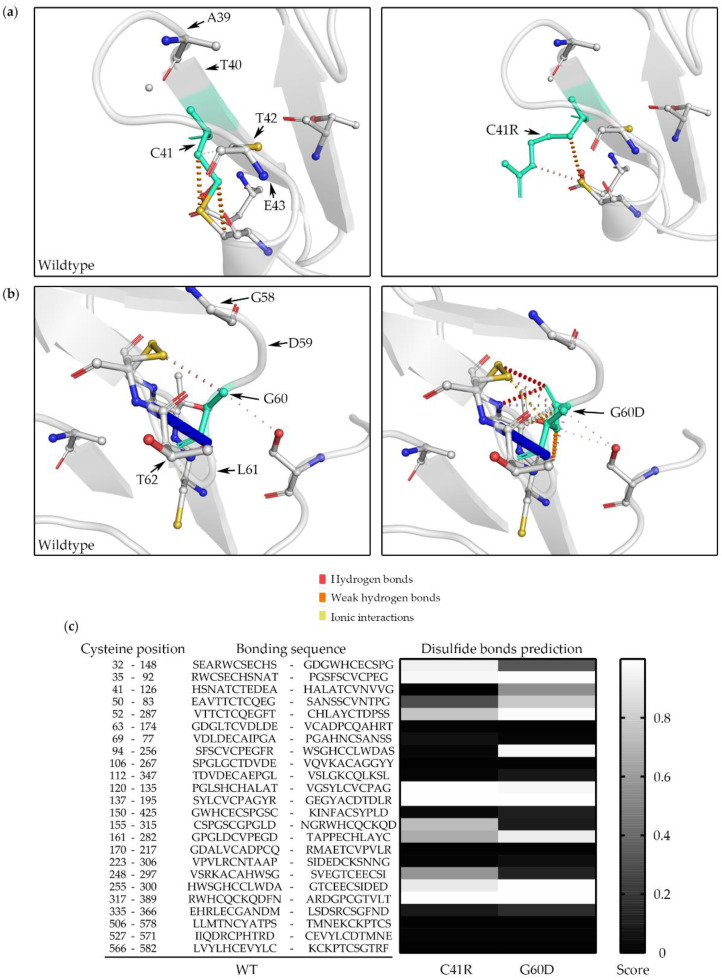
Uromodulin 3D structure and disulfide bond prediction. WT versus C41R (**a**) and WT versus G60D (**b**) were processed using DynaMut software. WT and mutant residues are colored in light green and are represented by sticks alongside the surrounding residues, which are involved in hydrogen bonds, weak hydrogen bonds, and ionic interactions. (**c**) Disulfide bond prediction of G60D and C41R uromodulin was analyzed using DiANNA software. The 24 disulfide bond positions and sequences of WT uromodulin are listed, and the prediction model shows that multiple disulfide bonds might be affected by C41R and G60D variants. WT: wildtype; G60D: p.Gly60Asp; C41R: p.Cys41Arg.

**Table 1 biomedicines-10-02265-t001:** Characteristics of *UMOD* variants in CKD individuals.

Family	Nucleotide Change	Amino Acid Change	ACMG Classification	Final Verdict	Clin Var	In Vitro Function	dbSNP	Allele Frequency
WG	c.121T > C	p.Cys41Arg	LP ^1^	LP		Delayed maturation and decreased extracellular excretion		
DY5	c.179G > A	p.Gly60Asp	VUS ^2^	LP		Delayed maturation and decreased extracellular excretion		
CKD401	c.817G > T	p.Val273Phe	VUS ^3^	LP	LP	Not performed	rs121917774	0.00006 (1/16760, 8.3KJPN)

Abbreviation: LP, likely pathogenic; VUS, variant of uncertain significance. ^1^ PM1: moderate, PM2: supporting, PP1: supporting, PP2: supporting, PP3: supporting. ^2^ PM1: supporting, PM2: supporting, PP1: supporting, PP2: supporting, PP3: supporting. ^3^ PM1: supporting, PM2: supporting, PP1: supporting, PP2: supporting, PP3: supporting.

## Data Availability

The data presented in this study are openly available in FigShare at: https://doi.org/10.6084/m9.figshare.20315697.v2 and https://doi.org/10.6084/m9.figshare.20317428.v1; accessed on 15 July 2022.

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
