# Peer review of "UMOD Mutations in Chronic Kidney Disease in Taiwan"

_biomedicines, 2022, doi:10.3390/biomedicines10092265_

Round 1
Reviewer 1 Report
this study can be published as set forth
Author Response
We appreciated the reviewer’s opinion and are glad the reviewer agreed with our statement.
Reviewer 2 Report
Huan-Da et al. examined the status of UMOD mutations in a selected CKD cohort in a single tertiary center in Taiwan, they concluded that UMOD mutations contributed to small but significant numbers of young individuals in Taiwan CKD. The reviewer has some concerns that need to be well addressed.
1. The MS indicated that uromodulin can be cleaved and excreted outside the cells. What is the difference between full-length-uromodulin and cleaved-uromodulin? Is there any method to detect these forms, separately? In this way, the uromodulin expression profile presented in figure 3, is it the full length-uromodulin or cleaved-uromodulin? What is the size? If it is the full-length-uromodulin, how about the cleaved-uromodulin?
Figure 3b, the levels of uromodulin in the culture medium should be normalized by the amount of protein. And can it further be verified by using WB?
In figure 3c, the levels of uromodulin in the cytosol need to be detected.
2. p.Gly60Asp and p.Cys41Arg, are they located in the cleavage site, thus affecting the cleavage of uromodulin? Or do they just affect the trafficking of uromodulin?
3. Do these UMOD Mutations contribute to kidney injury? The relevance and importance need to be assessed. Thus, the health status of cells after transient transfection needs to be evaluated.
Reviewer 3 Report
Thank you to the authors for submitting this manuscript describing assessment of a Taiwanese CKD cohort for ADTKD-UMOD. I have several concerns:
1) Abstract line 27-28, You should be referring to recent studying "showed that ADTKD-UMOD" rather than just "UMOD".
2) Abstract line 30, please use full and standard variant annotation (c., p., transcript)
3) Abstract, Is your denominator the number of families or individuals recruited? Please be much more specific.
4) Abstract line 35, again you are referring to "ADTKD-UMOD" rather than just "UMOD".
5) Line 57, Should read "UMOD mutations is delayed maturation, etc"
6) Please replace ESKD with Kidney Failure (KF) to confirm with current nomenclature as per KDIGO recommendations
7) Line 68, remove the word "mutation"
8) Line 69, change the word "status" to prevalence, and also change "UMOD" to "ADTKD-UMOD".
9) What is referred to by "DY5" families? Who was actually included? Are all 220 patients unrelated or are there 221 probands included with one of then belonging to family DY5?? This section is very confusing.
10) Line 75, is it also an inclusion to have an eGFR <90 before 40years of ago, or at any age? Please be more specific.
11) lines 101-102, these two sentences are confusing. Did all patients included in the study undergo exome sequencing and UMOD Exon 2/3/4/5 sanger sequencing?
12) The results are presented poorly and in a very confusing way. Please start with your topline and whole of study results and then delve into individual families/outcomes.
13) Please give all specific ACMG criteria which collectively result in the ACMG classification of each variant. This MUST resolve whether they are asserted as a VUS or LP/Path variant.
14) Line 269, this sentence makes no sense. Please rewrite and be clearer.
15) line 272-273, this sentence makes no sense. Please rewrite and be clearer.
16) Line 281, the use of VUS and pathogenic is used in far too casual a manner. Please be clearer, more structured and more deliberate who your flow of statements.
17) line 311, please continue to use standard variant nomenclature throughout.
18) Line 317-318, please clarify what is meant by this? Did all patients undergo exome sequencing?
Round 2
Reviewer 2 Report
I have no further comments.
Author Response
We appreciated the reviewer’s opinion and are glad the reviewer agreed with the reply in the revised manuscript.
Reviewer 3 Report
Thank you for submitting this revised manuscript. I have further queries:
1) Line 103, Should this read 220 CKD families? DY5 is in addition to that.
2) Table 1, please include a final variant classification according to ACMG criteria which incorporates the outcomes of your functional experiments. Also please add a column with the high level outcome/s of your functional experiments.
3) Results, please give a succinct statement somewhere that summarises clearly the outcomes of your functional experiments. Do they or do they not support pathogenicity for each individual variant?
4) The use of "mutation" throughout remains far too casual and inappropriate. Change to "variant" wherever possible.
5) Restrict your assertion of a clear diagnostic outcome ONLY to those cases harbouring a LP/Path variant.
Round 3
Reviewer 3 Report
Thank you for the further revisions. I have no further queries.